# Immunostimulatory Effects of Polysaccharides from *Spirulina platensis* In Vivo and Vitro and Their Activation Mechanism on RAW246.7 Macrophages

**DOI:** 10.3390/md18110538

**Published:** 2020-10-28

**Authors:** Xueyan Wu, Zhicong Liu, Yang Liu, Yu Yang, Fulin Shi, Kit-Leong Cheong, Bo Teng

**Affiliations:** Department of Biology & Guangdong Provincial Key Laboratory of Marine Biotechnology, Institute of Marine Sciences, College of Science, Shantou University, Shantou 515063, China; 14xywu1@stu.edu.cn (X.W.); 17zcliu2@stu.edu.cn (Z.L.); 17yyang2@alumni.stu.edu.cn (Y.Y.); 16flshi@alumni.stu.edu.cn (F.S.); klcheong@stu.edu.cn (K.-L.C.); bteng@stu.edu.cn (B.T.)

**Keywords:** *Spirulina platensis* (S.p.), polysaccharide, immunostimulatory, effects, mechanism

## Abstract

In this study, *Spirulina platensis* (S.p.) polysaccharide (PSP) was obtained by ultrasonic-microwave-assisted extraction (UMAE) and purified by an aqueous two-phase system (ATPS). Two different methods were applied to purified *Spirulina platensis* (S.p.) polysaccharide (PSP), respectively, due to PSP as a complex multi-component system. Three polysaccharide fractions (PSP-1, PSP-2, and PSP-3) with different acidic groups were obtained after PSP was fractionated by the diethyl aminoethyl (DEAE)-52 cellulose chromatography, and two polysaccharide fractions (PSP-L and PSP-H) with different molecular weight were obtained by ultrafiltration centrifugation. The chemoprotective effects of PSP in cyclophosphamide (Cy) treated mice were investigated. The results showed that PSP could significantly increase spleen and thymus index, peripheral white blood cells (PWBC), and peripheral blood lymphocytes (PBL). The in vivo immunostimulatory assays demonstrated that PSP could in dose-dependent increase of TNF-α, IL-10, and IFN-γ production in sera. The in vitro immunostimulatory assays showed that PSP and its fractions (PSPs) could evidently enhance the proliferation of splenocytes and RAW 264.7 cells and increase the productions of nitric oxide (NO), tumor necrosis factor-α (TNF-α), and interleukin 6 (IL-6). PSPs could also enhance phagocytic activity of RAW 264.7 cells. The acidic polysaccharide fractions of PSP-2, PSP-3, and PSP-L with small molecular weight had the higher immunostimulatory activity. Signaling pathway research results indicated that PSP-L activated RAW264.7 cells through MAPKs, NF-κB signaling pathways via TLR4 receptor.

## 1. Introduction

Cancer is one of the leading causes of death in the world every year, and the second leading cause of death in developing countries [1]. One of the most frequently used therapeutic modalities for the treatment of cancer is chemotherapy, most of which can bring an impairment of the host defense mechanisms, leading to immunosuppressive and cytotoxic effects. Cyclophosphamide (Cy) is the most widely used anticancer agent in chemotherapy with a high therapeutic index and broad spectrum of activity against a variety of cancers [2]. Cy is an effective alkylating agent and able to bind to the DNA through covalent linkages (alkylation), particularly to the N7 of guanine residue, and forms DNA double strand adducts. However, they can also damage healthy cells and lead to multiple organs damage and immunosuppression, which limit their use in tumor treatment [3,4].

For recent years, many immunomodifiers composed of polysaccharide have been widely developed, such as lentinan, algin, krestin, and tea polysaccharides [5,6,7,8]. S.p. is known to be a member of the blue–green alga family, which has great immunoregulatory activity [9,10,11,12]. However, there is very few reports [13,14] about the immunomodulatory of its polysaccharide, and they were limited in cellular level, and also the structure of PSP was complex, and the structure-activity relationship between its structure and immunostimulatory activity was unclear. So, a comprehensive and in-depth study on the immunoregulatory activity of PSP appears to be lacking.

Macrophages was widely used in immunology experiments for they could initiate the innate immune response and induce the adaptive immune response in response to microbial infection, cancers, and immunological diseases [15,16]. Therefore, macrophage RAW246.7 cell line had been selected to study the immunostimulatory and the regulatory mechanism of PSP.

In this study, different PSPs which had different structural characteristics were fractionated by the DEAE-52 cellulose chromatography and ultrafiltration centrifugation, to further reveal the structure–activity relationship of PSPs, the immunostimulatory effect of PSP was tested in mice immunosuppressed by treatment with Cy and in RAW246.7 cell in vitro. To investigate the molecular mechanisms responsible for the polysaccharide-induced immunostimulating response, the receptor TLR4 and roles of signaling molecules such as nuclear factor (NF-κB) and mitogen-activated protein kinases (MAPKs) were further evaluated. 

## 2. Results and Discussion 

### 2.1. Effect of PSP on Spleen and Thymus Indices in Cy-Treated Mice

The spleen and thymus indices were examined to evaluate the effects of PSP on the immune organs. As was shown in Table 1, the thymus and spleen indices were significantly decreased in the Cy-treated group compared with the NC (normal control) group (*p* < 0.01). The thymus and spleen indices of the mice treated with PSP of different doses increased in a dose-dependent manner as compared with the model control. Lentinan (LNT)-treated (500 mg/kg BW/day) also remarkably increased the spleen indices of the mice compared with model control (*p*< 0.01). The results suggested that cyclophosphamide could sharply reduce the thymus and spleen index, and high dose PSP (1500 mg/kg BW/day) could effectively protect the thymus and spleen index decreased significantly. 

### 2.2. Effect of PSP on Hemopoietic Function in Cy-Treated Mice

To evaluate the protective effect of PSP on the myelosuppression induced by Cy, the white blood cells and lymphocytes from peripheral blood in Cy-treated mice were counted. It was found that PWBC and PBL counts in Cy-treated mice decreased significantly (Table 1). The PWBC and PBL counts of the PSP-treated and LNT-treated group were elevated remarkably compared with the model group, and PSP-treated groups did in a dose-dependent manner. These results indicated that PSP could alleviate the myelosuppression induced by Cy.

### 2.3. Effect of PSP on Cytokine Levels in Sera from Cy-Treated Mice

As shown in Figure 1, Cy injection caused significant reduction in the levels of TNF-α, IL-10, and IFN-γ. After a low, middle, or high dose of PSP administration, there were a dose-dependent increase of TNF-α, IL-10, and IFN-γ production in sera compared to the model control group. Significant increases in concentrations of the cytokines (TNF-α, IL-10, and IFN-γ) were also observed in the LNT-treated group when compared with the Cy-treated group (*p* < 0.05 or *p* < 0.01). The result illustrated that PSP treatment on cytokine levels counteracted the immunosuppressive effect caused by Cy.

### 2.4. Effect of PSPs on Cell Proliferation 

The effects of PSPs on mouse spleen lymphocyte and RAW 264.7 cells proliferation were determined by CCK-8 assay. As shown in Figure 2 and Figure 3, the results showed that PSPs were nontoxic to spleen lymphocyte or RAW264.7 cells within tested concentrations. On the contrary, PSPs could promote cell proliferation of spleen lymphocyte and RAW264.7 cells, especially the low molecular polysaccharide PSP (PSP-L) had a significant effect to promote cell proliferation in both spleen lymphocyte and RAW264.7 cells (*p* < 0.01). Among PSP-1, PSP-2, and PSP-3, the acid polysaccharide PSP-2 acted as a better proliferation stimulus of RAW264.7 cells.

### 2.5. Effect of PSPs on the Production of NO, IL-6, and TNF-α by RAW 264.7 Cells

In the present study, the effect of PSPs on macrophages activation was investigated via measuring the production of NO, IL-6, and TNF-α in culture medium. The results (Figure 4, Figure 5 and Figure 6) showed that all PSPs exhibited good ability to promote NO, IL-6, and TNF-α production of RAW264.7 cells. PSPs increased the NO secretion in a concentration-dependent manner (50–2000 μg/mL). All PSPs also induced cytokines secretion, and increased NO in a concentration-dependent manner after 24 or 48 h, but only PSP-H and PSP-L increased NO, IL-6, and TNF-α in a concentration-dependent manner. When RAW264.7 cells were stimulated by 2000 μg/mL PSP, the highest production of NO was tested, and the production of NO and IL-6 increased higher than the LPS (lipopolysaccharide)-treated group. The acid polysaccharide PSP-2 showed higher activity in promoting cytokine secretion among PSP-1, PSP-2, and PSP-3. PSP-L displayed a time dependency.

### 2.6. Effects of PSPs on Phagocytic Activity of RAW264.7 Cells

A distinguished feature of activated macrophages is illustrated by an increase in phagocytosis [17], which in turn demonstrates the activation of the innate immune response. Neutral red assay was employed here, in order to evaluate the effects of PSPs on the phagocytic activity of RAW264.7 cells. As showed in Figure 7, after 24 or 48 h incubation, the phagocytic OD values of RAW264.7 of the LPS group and PSPs group increased in varying degrees compared with the control group, indicating that the phagocytic activity of RAW264.7 cells had been enhanced by LPS and PSPs. However, under the low concentration of PSPs, the effects of PSPs were not obvious. As the concentration increases, the phagocytic activity was significantly enhanced. In the case of PSPs after incubation 24 h, the phagocytic OD values of RAW264.7 increased in a dose-dependent manner except PSP-1, and after incubation 48 h, the OD values increased in a dose-dependent manner except PSP. Among PSP and its fractions, PSP-L had the most obvious effect in promoting phagocytic activity of RAW264.7 cells. Such an enhancement of phagocytic activity suggests that PSPs were capable of inducing macrophage activation.

### 2.7. Effects of TLR4 Inhibitor on the Cytokines Secretion by RAW264.7 Cells

In order to investigate the role of TLR4 played in PSP-mediated stimulation of macrophages, TAK-242 (TLR4 inhibitor) was pre-incubated with RAW264.7 cells for 1 h and then incubated with PSP-L or LPS. The production of TNF-α and IL-6 in RAW264.7 cells was measured. As shown in Figure 8 and Figure 9, the levels of IL-6 and TNF-α were sharply decreased in comparison to the group treated with PSP-L only (*p* < 0.01). This result suggested that TLR4 was one of the receptors of PSP-L on RAW264.7 cells and may be the vital one.

### 2.8. Western Blot Analysis

To further unravel the mechanisms underlying the activation of RAW264.7 cells, Western blot was also employed to assess the protein expressions of MAPKs (p38, ERK, and JNK). As shown in Figure 10, after the stimulation, the levels of phosphorylated p38, JNK, and ERK in the PSP-L-treated cells were detected higher than those in the control group. As the positive control, LPS stimulation also significantly increased phosphorylation levels of the three MAPKs. With the above results, it was shown that PSP-L, liked LPS stimulation, also activated the immunostimulating response of RAW246.7 macrophages by the MAPK signaling pathway. 

### 2.9. Effects of PSP-L on the Activation of NF-κB Signaling Pathway

With regard to the NF-κB signaling pathway, our results showed that with stimulation of LPS and PSP-L, IκB-α degraded and resulted in the release of NF-κB, and then the p65 subunit translocated from the cytoplasm to the nucleus. As shown in Figure 11, the red fluorescence of protein p65 was weak and only a small amount of p65 subunit in the nucleus. However, in the LPS and PSP-L treated group, the red fluorescence of protein p65 was bright and was concentrated in distribution. The picture of merging showed that a lot of p65 subunit was situated in the nucleus. These results indicated that PSP-L could induce the NF-κB nuclear translocation in RAW 264.7 cells.

## 3. Discussion

As is well known that Cy is an important chemotherapeutic drug in tumor treatment to inhibit the proliferation of cancer cells. Cy typically causes side effects such as myelosuppression and immunosuppression by damaging DNA of normal cells [18]. Many polysaccharides isolated from plants, fungi, algae, and animals have proven to alleviate Cy-induced side effects because of their immunostimulating activities [4]. Cy-induced immunosuppression in laboratory animals was considered as a relevant model for investigating the immunomodulatory potential of dietary components [19]. In this study, Cy-treated mice were used as a model group to evaluate the immunostimulatory activity of PSP in vivo. As expected, Cy markedly reduced PWBC and PBL counts, lowered the spleen and thymus index, and decreased the levels of cytokines (IL-10, TNF-α, and INF-γ) in sera. These data proved immunosuppressed mice were successfully modeled and were consistent with previous reports [20,21].

Immune function and immune prognosis are reflected well by the thymus and spleen indices because these organs play important roles in specific and nonspecific immunity. A healthy spleen provides the body with anti-inflammatory, anti-tumor cytokines, natural killer cells, and different types of Th cells, which help the body avoid the infection of various bacteria and virus. Thymus is the central immune organ of the body, where T cell differentiation, development, and maturation occur. When immune function strengthens, immune organs’ weight increases because of immune cell proliferation and differentiation. Similarly, immune organs’ weight lower when immune function decreases [22]. In this study, significant increases in the thymus and spleen indices were observed in the PSP/LNT-treated groups compared with the Cy-treated group. The increase in the spleen index of mice treated with PSP indicated that PSP was able to survive against the immunosuppression induced by Cy.

Bone marrow is an important central immune organ of the body. The generation of many kinds of immune cells and the differentiation and maturation of B cells are completed in bone marrow. Myelosuppression is an important limiting factor on the outcome and recovery of tumor patients receiving chemotherapy. WBC (white blood cells) and BL (blood lymphocytes) counts reduction reflected the hematopoietic stem cells damage caused by Cy [23]. The findings showed administration of PSP restored PWBC and PBL counts in a dose-dependent manner, suggesting that PSP could provide protection against myelosuppression induced by Cy.

The different cytokine patterns secreted by Th1 and Th2 cells are major determinants of the differences in cellular function [23]. IL-2, IFN-γ, and TNF-α were secreted by Th1 cells, which mainly mediate cellular immune response, whereas Th2 cells secrete IL-4, IL-6, and IL-10, promoting the humoral immune response [24,25]. Different cytokines often act together and affect the synthesis of other cytokines. In this study, IL-10, IFN-γ, and TNF-α were measured to evaluate the effect of PSP on Th1/Th2. Based on the results, it showed that PSP treatment accelerated recovery of the level of IL-10, IFN-γ, and TNF-α in the serum in Cy-treated mice, suggesting that PSP may enhance cellular and humoral immune function by increasing the secretion of the three cytokines. 

As we all know, macrophage activation is accepted as one of the most important events in the immune response and macrophage activation signifies the up-regulation of the innate immune response [26]. The immune activity of PSPs in vitro was measured by proliferation of RAW246.7 cell and splenic lymphocyte, the production of NO and cytokines in RAW264.7 cells culture medium and their phagocytosis. As a messenger molecule and a cytotoxic molecule, NO is widely involved in various physiological processes related to the immune response. NO is believed to be a major mediator of macrophages and essential for the resistance of immune system to pathogens invasion [27]. The cytokines TNF-α and IL-6 can induce tumor cell apoptosis and result in tumor necrosis, acting as critical roles in mediating the signal transduction which stimulates the immune defense system [28]. All PSPs exhibited good ability to promote NO, IL-6, and TNF-α production, the NO and IL-6 secretion levels of the PSP-treated cells (at 2000 μg/mL) were detected higher than those in LPS-treated group, respectively, indicating that PSPs exhibited remarkable immunostimulatory activity.

Among PSP-1, PSP-2, and PSP-3, the acid polysaccharide PSP-2 played a higher immunostimulatory activity by promoting more NO and cytokines secretion. The capability of PSP-2 to activate macrophage cells was thought to be governed by the amount of sulfate groups presented in their structure [29]. Some polysaccharide research found that the lower molecular weight, polysaccharides more significantly induced proinflammatory response in macrophage cells [30]. This was consisted with our finding that PSP-L promoted more proliferation in both RAW264.7 cells and spleen lymphocyte.

PSP-L was chosen to study the mechanism of PSP on RAW246.7 cell activation because of its better dose-dependent stimulating. The immunomodulatory activity of polysaccharides is based on the stimulation of pattern recognition receptors [31]. TLR4 predominately recognizes LPS, lipoteichoic acid, and plant polysaccharides and is one of the most studied receptors in immune activity [32]. Based on the results, we concluded that TLR4 was one of the membrane receptors of PSP-L on RAW246.7 cell. NF-κB plays an important role in the proliferation, differentiation, and activation of macrophages. Previous research has shown that TLR4 could activate the NF-κB signal pathway by a large number of inducers [32]. In this study, findings seem to indicate that PSP-L plays an immune-stimulatory role in macrophages via TLR4-induced activation of the NF-κB signal pathway. Meanwhile, the increased phosphorylation levels of the three MAPKs were also observed. They suggested that the potential of PSP-L as an immunostimulatory agent by increasing the secretion of NO, IL-6, and TNF-α in RAW264.7 cells, which may be via increasing the mRNA transcription by p38/JNK/ERK pathways. These results demonstrated that activation of NF-κB and MAPKs were all involved in the process of PSP-L mediated activation in macrophages, as shown in Figure 12.

By comparing the immunostimulatory effects of PSPs in vitro, it was found that acidic polysaccharides PSP-2 and PSP-3 (concentration-dependent) had better effects on proliferation of RAW264.7 cells and cytokines secretion than neutral polysaccharide component PSP-1, which may be related to the more sulfate group and uronic acid group contained in PSP-2 and PSP-3, as shown in Figure 13. The result was consistent with other acidic polysaccharide research [33]. From the proliferation of RAW264.7 cells experiment results, it could be observed that the small molecular weight (under 10 kDa) polysaccharides show outstanding proliferation activity compared with other fractions of polysaccharides after 48 h, suggesting that the molecular weight is an important factor affecting the immunostimulatory effects of PSPs. Based on the comprehensive evaluation of PSPs from the two aspects of acid group and molecular weight, they hypothesized that fraction of PSP, which has the small molecular weight and acidic polysaccharide, has the highest immunostimulatory activity, as shown in the green area of Figure 13.

## 4. Materials and methods

### 4.1. Reagent

S.p. was provided by the Here Biotechnology Company (Yunnan, China) in form of powder. DEAE cellular 52, cytokines enzyme-linked immunosorbent assay (ELISA) kits were provided by Beijing Solarbio Biotechnology Company (Beijing, China). Rabbit polyclonal GAPDH antibody, rabbit monoclonal p-ERK1/2, ERK1/2, p-JNK, JNK, p38, p-p38 antibody were purchased from Abcam (Shanghai, China). NF-κB nuclear transfer kit, detection kit, were purchased from Beyotime Biotechnology (Shanghai, China). TAK-242 (TLR4 inhibitor) was provided by Abmole (Shanghai, China). Cy was purchased from Shanxi Pude Pharmaceutical Co. LTD. (Datong, China) Lentinan (LNT) was purchased from Wuhan Yuancheng Gongchuang Technology Co. Ltd. (Wuhan, China), the purity is about 53.6%. Other chemical regents were of analytical grade, and were obtained from Xilong Scientific Company (Guangdong, China). Aqueous solutions were prepared with deionized and doubly distilled water.

### 4.2. Preparation and Quantitative Analysis of PSP

PSP was isolated and purified as reported [34] with modifications. PSP was extracted by UMAE using an ultrasonic-microwave apparatus (CW-2000, Shanghai Xintuo Microwave Instrument Co. Ltd., Shanghai, China) at a microwave power of 800 W and an ultrasonic power of 50 W and the optimized extraction condition was 19.31 min, 75.16 °C, and solid–liquid ratio of 1.0:92.35. The extracts were centrifuged, and the supernatant was collected. After concentration, ethanol was added to the final concentration of 80% (*v*/*v*) for precipitation of polysaccharides. This crude extract was further purified by ethanol-ammonium sulfate ATPS to remove protein at TLL = 50 (TLL is tie line length of ATPS) [35]. After the phase separation, the bottom phase solution was dialyzed and freeze-dried in vacuum to obtain PSP.

The PSP were loaded onto a DEAE-52 cellulose column chromatography, which was then eluted at a flow rate of 0.5 mL/min successively with distilled water, 0.4 mol/L NaCl, and 0.7 mol/L NaCl solution. After that, these 3 PSP fractions was dialyzed and freeze-dried in vacuum to obtain PSP-1, PSP-2, and PSP-3, whose structural analyses were performed and displayed in the Appendix A. Meanwhile, PSP was partitioned by 10 kDa tubular ultrafiltration membrane to get PSP-H, of which molecular weight is higher than 10 kDa, and PSP-L with molecular weight less than 10 kDa.

### 4.3. Animals and Experimental Design

Male Kunming mice (4-weeks-old, 22 ± 2 g) were obtained from Hunan SJA Laboratory Animal Co., LTD. All experiments were performed following the protocol for animal study approved by the Ethics Committee of Medical College of Shantou University (SUMC2020-227, 12 May 2020). All experimental procedures involving the handling and caring of animals have been carried out in accordance with the ethical guidelines. All animals were housed and maintained on a standard commercial diet at ambient temperature in a clean environment as mentioned in the ethical guidelines. The temperature was maintained at 25 ± 2 °C and relative humidity at 60 ± 10% with a 12 h light/dark cycle.

The mice were randomly divided into 6 groups consisting of 12 mice each. All animals were allowed 5 days to adapt to their environment before the treatment. One group of healthy mice was used as the normal control (NC) group and treated once daily with normal saline (NS) for 14 days. From day 2, the other five groups of mice were subjected to immunosuppression by administration of Cy (30 mg/kg/day) intraperitoneally every other day. From day 1, the mice were administered the following (Table 2): NC group, NS; Cy group, normal saline; PSP_L_, PSP_M_, and PSP_H_ groups (PSP_L_, PSP_M_, and PSP_H_ denoted low, middle, and hight dose of PSP), 500, 1000, or 1500 mg/kg/day body weight PSP; and positive control group, 500 mg/kg/day bodyweight LNT (lentinan had been illustrated that it had good therapeutic effect on the immunosuppression mice [36,37]). All animals were orally administrated daily. The schematic diagram of immunosuppressed mice molding and treating process is shown in Figure 14.

### 4.4. Calculation of Splenic and Thymic Indices

Twenty-four hours after the last drug administration, 6 mice of each group were weighed and sacrificed by cervical dislocation. The spleen and thymus were immediately removed and weighed. The visceral index was calculated as visceral weight (mg)/body weight (g).

### 4.5. Peripheral White Blood Cell and Peripheral Blood Lymphocytes Cells Counts

Blood was collected from 6 mice above, and into blood routine tubes on the day of sacrifice by enucleating the eyeball. Peripheral white blood cell (PWBC) and peripheral blood lymphocytes (PBL) were analyzed using an automatic hematology analyzer (SYSMEX XN-2800, Sysmex Corporation, Shanghai, China).

### 4.6. Determination of IL-10, INF-γ, and TNF-α in Serum

Serum was collected from other 6 mice of each group by enucleating the eyeball 24 h after the last administration of PSP. The concentrations of IL-10, INF-γ, and TNF-α in the sera were determined using ELISA kits according to the instruction of the manufacturer.

### 4.7. Cell Culture

The RAW 264.7 murine macrophages were obtained from the Cell Bank of Type Culture Collection of Chinese Academy of Sciences (Shanghai, China) and cultured in Dulbecco’s modified Eagle’s medium (DMEM), supplemented with 10% inactivated fetal bovine serum and 1% penicillin-treptomycin solution. Cultures were maintained at 37 °C in a humidified 5% CO_2_ incubator.

### 4.8. Assay for Cell Viability

After RAW 264.7 cells were collected and suspended in DMEM, the cell density was adjusted to 1 × 10^6^ cells/mL, a total of 100 μL cell suspension was added in the 96-well flat-bottom plate and incubated for 12 h at 37 °C in a humidified incubator with 5% CO_2_. Adherent RAW 264.7 cells were treated with increasing concentrations of PSPs (10–2000 μg/mL) or lipopolysaccharide (LPS) 10 μg/mL, in the growth medium at 37 °C in 96-well plates. After 24 h or 48 h incubation, 10 μL Cell Counting Kit-8 (CCK-8) was added to each well and incubated for additional 30 min. Absorbance in each well was measured at 450 nm using a microplate reader (BioTeK Instruments, Inc., Winooski, VT, USA). The cell viability (%) was calculated as following:Cell viability (%) = A_S_/A_0_ × 100%,(1)
where A_S_ is the absorption value of the experimental group and A_0_ is the light absorbance value of the blank control group, respectively.

### 4.9. Spleen Lymphocyte Proliferation Assay

Spleen collected from sacrificed healthy mice under aseptic conditions were chopped into small pieces and passed through a fine steel mesh to obtain a homogeneous cell suspension. Recovered spleen cells were resuspended in red blood cell lysis buffer for 5 min to remove erythrocytes. Then spleen cells were harvested and resuspended in DMEM medium. After centrifugation (200 g, 5 min), the cells were resuspended to a concentration of 2 × 10^6^ cells/mL in DMEM medium. Then, the same volume of PSPs or LPS solution was added to a final concentration 50, 500, 2000 μg/mL or 10 μg/mL. After 72 h incubation in 96 well-plates, the above CCK-8 method was used to measure the cell viability. 

### 4.10. Measurement of NO, TNF-α, IL-6 Cytokine Production

After being cultured for 24 h in 96-well plate (1 × 10^6^ cells/well) at 37 °C in a humidified incubator, DMEM medium in the presence or absence of PSPs (50, 500, 2000 μg/mL) or LPS (10 μg/mL) were added to RAW264.7 cells culture wells followed by incubation for another 24 or 48 h. Then, the culture medium containing NO and cytokines secreted from active macrophage RAW264.7 cells were collected for further analysis. The levels of NO and cytokines were measured using nitric oxide assay kit, TNF-α, and IL-6 ELISA kits, respectively. LPS was used as a positive control. The DMEM medium with no PSP was used as blank control. All pre-experiments were done to confirm that the dilution rate was appropriate for kit tests. 

### 4.11. Determination of Pinocytic Capability of RAW264.7 Cells

The determination of pinocytic capability was carried out as the previous research described with minor modifications [38]. RAW264.7 cells were collected during the logarithmic phase and seeded at 1 × 10^6^ cells/well in a 96-well plate and incubated at 37 °C in a humidified atmosphere with 5% CO_2_. After 24 h, DMEM medium, LPS (10 μg/mL), or PSPs (50, 500, 2000 μg/mL) were added into each well, and these cells were incubated at 37 °C for 24 or 48 h. Each concentration was repeated six wells. A total of 100 μL 0.09% neutral red was added to replace the old medium and incubated for 30 min. The plate was washed three times with PBS and 100 μL cell lysis solution (ethanol: acetic acid = 1:1 *v*/*v*) was added into each well. The absorbance was acquired at 540 nm after completely lysis. The pinocytic capability was calculated as following:Pinocytic capability (%) = (A_S_/A_0_) ×100%,(2)
where A_S_ and A_0_ were the absorbance of sample and the absorbance of the control, respectively.

### 4.12. Investigation of Membrane Receptors

RAW264.7 cells seeded at 1.5 × 10^6^ cells/well in a 48-well plate and incubated in a humidified incubator for 24 h. Remove the culture medium and washed the cell with PBS for 2 times. Then, the cells were pre-treated with 0.1% DMSO, 10 μmol/L TAK-242 (TLR4 inhibitor, dissolved in DMSO and diluted with DMEM medium) for 1 h, followed by treatment with DMEM medium PSP-L (50, 500, 2000 μg/mL) or LPS (10 μg/mL) for 24 h. Next, the culture solution was collected in a sterile centrifuge tube, centrifuged at 2000 rpm for 5 min at 4 °C, and the supernatant was collected. The levels of IL-6 and TNF-α in the culture supernatants were measured by mouse TNF-α ELISA kit, and mouse IL-6 ELISA kit, respectively, as described above.

### 4.13. Western Blot Analysis

Western blotting was performed according to standard procedures. RAW 264.7 cells were treated with PSP-L (50, 500, 2000 μg/mL) or LPS (10 μg/mL) for 30 min. After washing with cold PBS (3 times), cells were lysed with 100 μl cell lysis buffer mixed with 1% PMSF (100 mM). The protein contents were measured with the BCA protein assay kit using bovine serum albumin (BSA) as a standard. The extracts were mixed with SDS-sample buffer and incubated at 100 °C for 5 min. The denatured proteins were separated on 10% sodium dodecyl sulfate polyacrylamide gel electrophoresis (SDS-PAGE), and then were transferred to a polyvinylidene difluoride (PVDF) membrane. After being blocked with 5% skim milk in TBS-0.1% Tween 20 (TBST), the blot was incubated with target monoclonal antibody (p-JNK, JNK, p-ERK1/2, ERK1/2, p-p38, p38, GAPDH) in TBST containing 5% BSA overnight at 4 °C. Subsequently, the membranes were washed with TBST and incubated with corresponding horseradish peroxidase-conjugated secondary antibodies for 2 h. After washing the membrane with TBST three times for 10 min, protein bands were examined by enhanced chemiluminescence (ECL, Beyotime Biotechnology, Shanghai, China) and the optical density of blots were analyzed by Quantity One (Bio-Rad, San Francisco, CA, USA).

### 4.14. Immunofluorescence Staining Analysis of Nuclear Translocation of NF-κB

The cells (1 × 10^5^ cells/mL) were cultured with DMEM medium, PSP-L (2000 μg/mL) or LPS (10 μg/mL) for 24 h. The culture medium was discarded, and fixing liquid was added to the cells. After 10 min, cells were washed three times with PBS for 5 min each washing. Next, cells were incubated with the primary NF-κB p65 antibody at 4 °C overnight and then washed with a washing solution three times for 10 min each, followed by incubation with a Cy3-conjugated secondary antibody at room temperature for another 1 h, and then washed two times. 4’,6-diamidino-2-phenylindole (DAPI) was added for staining and then removed after 5 min by washing three times. Finally, samples were visualized under a fluorescent inverted microscope (37XC, Shanghai optical instrument factory, Shanghai, China). All steps were under the instructions of NF-κB nuclear transfer kit and detection kit.

### 4.15. Statistical Analysis

Each experiment was performed in triplicate to minimize deviation. The data, hence obtained, were presented as mean ± S.D. Data were analyzed one-way ANOVA using SPSS Statistics 20 software (Version 20, IBM Co., Armonk, NY, USA) and Dunnett’s test (Version 20, IBM Co., Armonk, NY, USA). P-values of less ± by one-way ANOVA using SPSS and Dunnett’s test. P-values of less than 0.05 were assumed to be statistically significant.

## 5. Conclusions

The present study confirmed that the polysaccharide fractions from Spirulina platensis (PSP) with different structure characteristics could alleviate cyclophosphamide-induced immunosuppression in mice by protecting spleen and thymus, increasing peripheral white blood cells (PWBC) and peripheral blood lymphocytes (PBL) and restored the levels of TNF-α, INF-γ, IL-10 in the sera. The in vitro assays proved its immunomodulatory activity and indicated that its fraction PSP-L could activated RAW264.7 cells through MAPKs, NF-κB signaling pathways via TLR4 receptor, but more control and other experiments are needed to be able to assume the involvement of TLR4 in the recognition of PSP-L. Moreover, the acidic polysaccharide fractions of PSP-2, PSP-3, and PSP-L with small molecular weight had the higher immunostimulatory activity.

## Figures and Tables

**Figure 1 marinedrugs-18-00538-f001:**
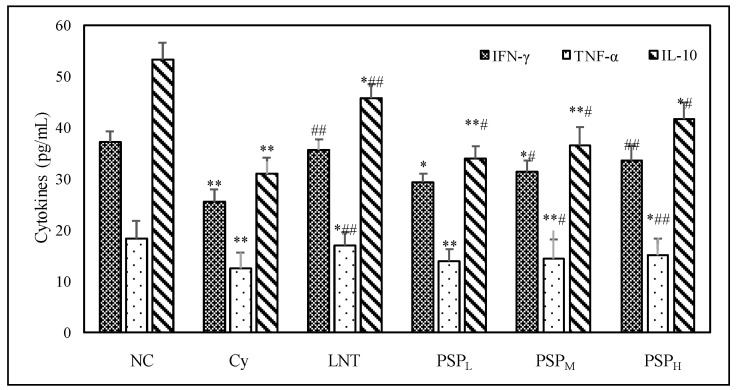
The effects of PSP on cytokines in serum of immunosuppressed mice. PSP_L_ denotes low dose PSP, PSP_M_ denotes middle dose PSP, PSP_H_ denotes high dose PSP, data shown are means ± SD (*n* = 6). All data were analyzed statistically using a one-way analysis of variance. (*****) *p* < 0.05 and (******) *p* < 0.01, compared with the normal control, and # *p* < 0.05 and ## *p* < 0.01 compared with the Cy group. respectively.

**Figure 2 marinedrugs-18-00538-f002:**
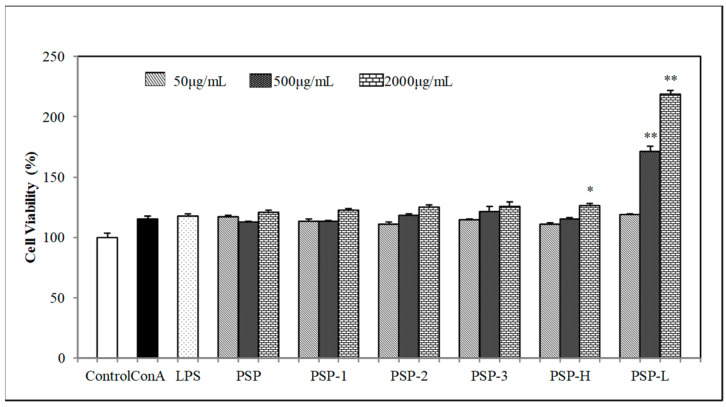
Effects of PSPs on mouse splenic lymphocyte proliferation. The data shown are means ± SD (*n* = 3). All data were analyzed statistically using a one-way analysis of variance. (*****) *p* < 0.05 and (******) *p* < 0.01, compared with the normal control, respectively.

**Figure 3 marinedrugs-18-00538-f003:**
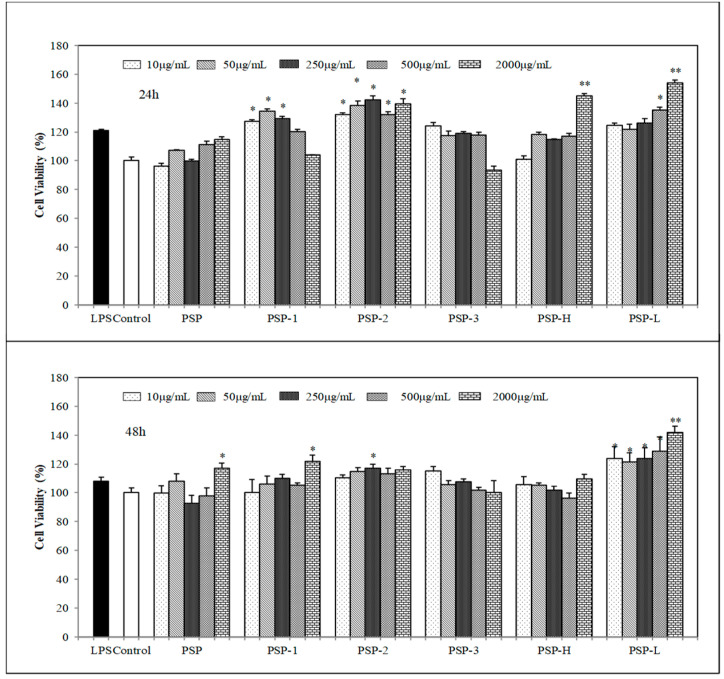
Effects of PSPs on cell viability of RAW264.7 cells. The data shown are means ± SD (*n* = 3). All data were analyzed statistically using a one-way analysis of variance. (*) *p* < 0.05 and (**) *p* < 0.01, compared with the normal control, respectively.

**Figure 4 marinedrugs-18-00538-f004:**
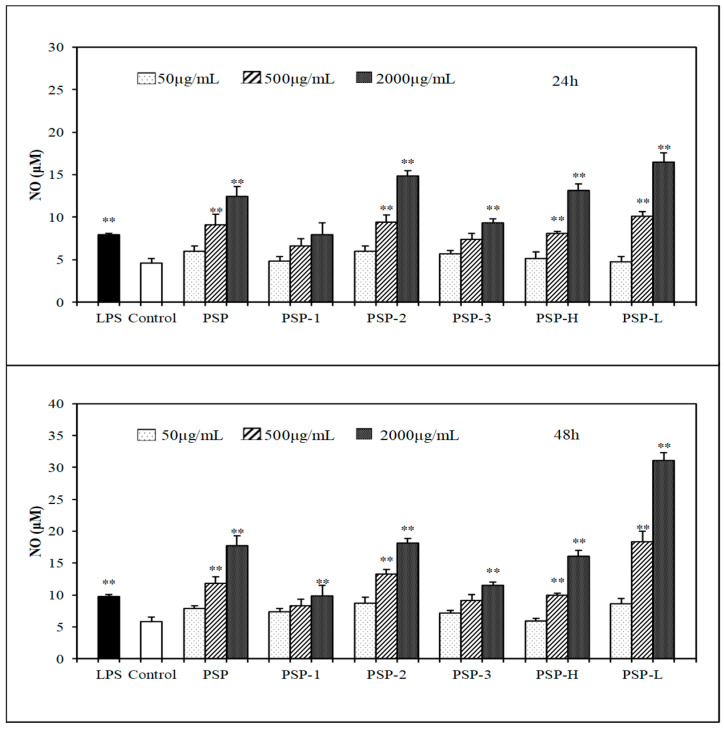
Effects of PSPs on inducing NO production of RAW 264.7 cells. The data shown are means ± SD (*n* = 3). All data were analyzed statistically using a one-way analysis of variance. (**) *p* < 0.01, compared with the normal control, respectively.

**Figure 5 marinedrugs-18-00538-f005:**
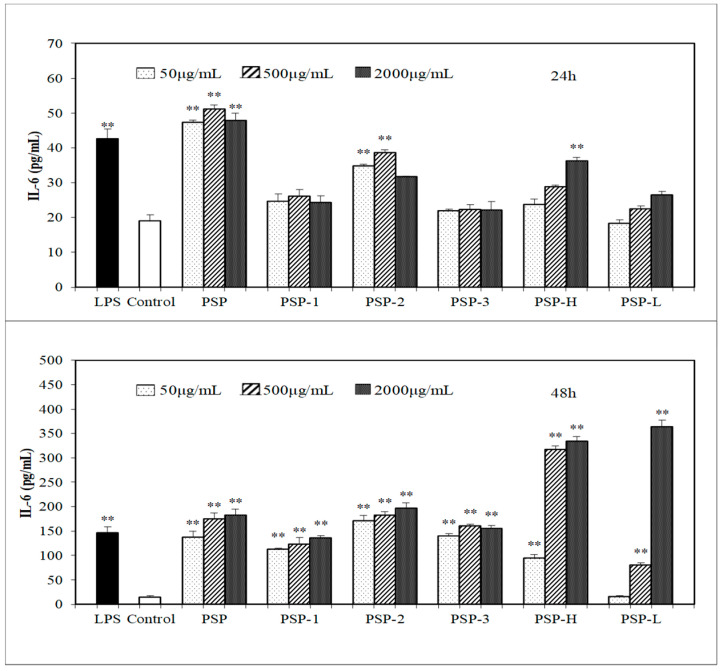
Effects of PSPs on inducing IL-6 production of RAW264.7 cells. The data shown are means ± SD (*n* = 3). All data were analyzed statistically using a one-way analysis of variance. (**) *p* < 0.01, compared with the normal control, respectively.

**Figure 6 marinedrugs-18-00538-f006:**
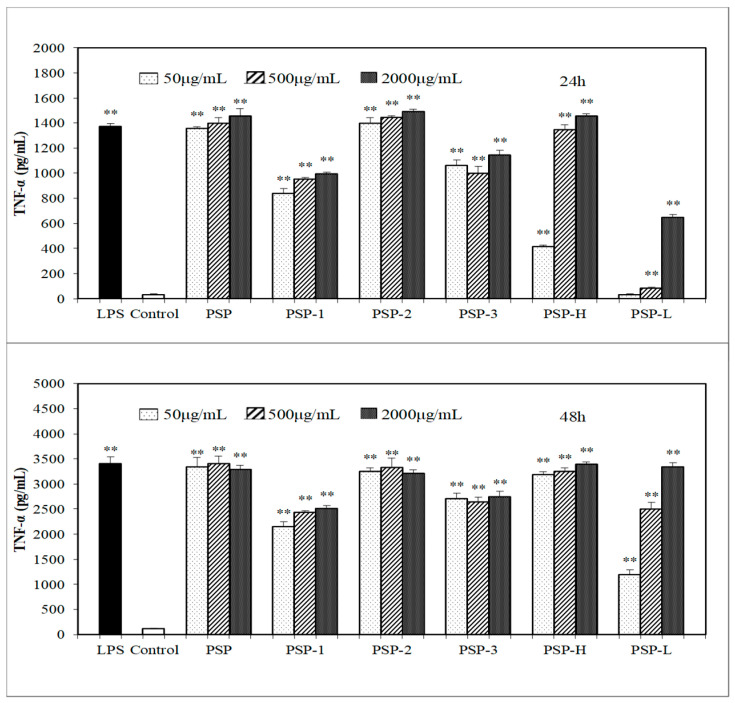
Effects of PSPs on inducing TNF-α production of RAW264.7 cells. The data shown are means ± SD (*n* = 3). All data were analyzed statistically using a one-way analysis of variance. (**) *p* < 0.01, compared with the normal control, respectively.

**Figure 7 marinedrugs-18-00538-f007:**
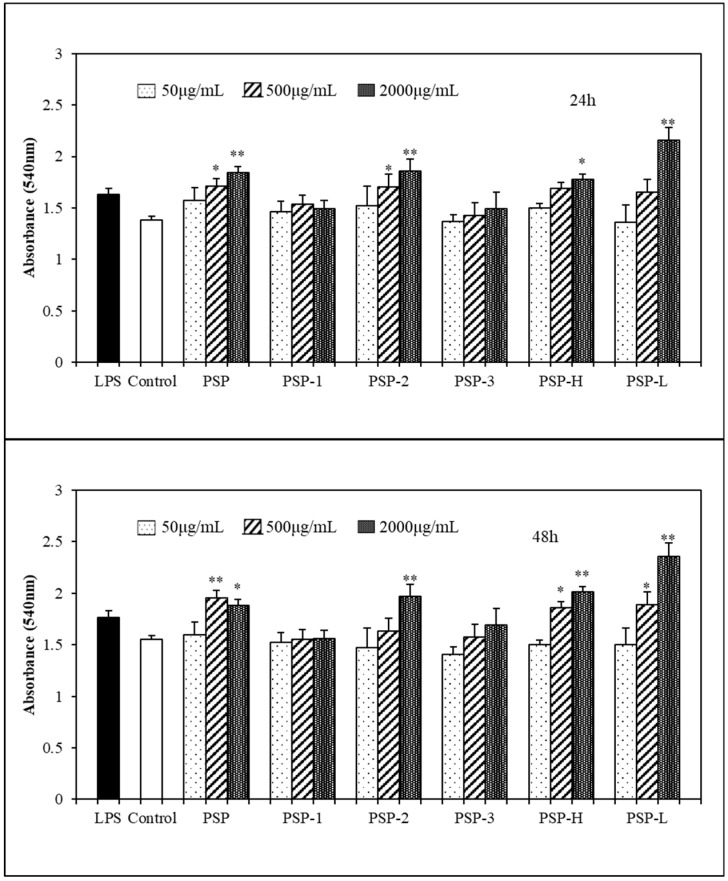
Effects of polysaccharides on the phagocytosis activity of RAW264.7 cells. The data shown are means ± SD (*n* = 3). All data were analyzed statistically using a one-way analysis of variance. (*) *p* < 0.05 and (**) *p* < 0.01, compared with the normal control, respectively.

**Figure 8 marinedrugs-18-00538-f008:**
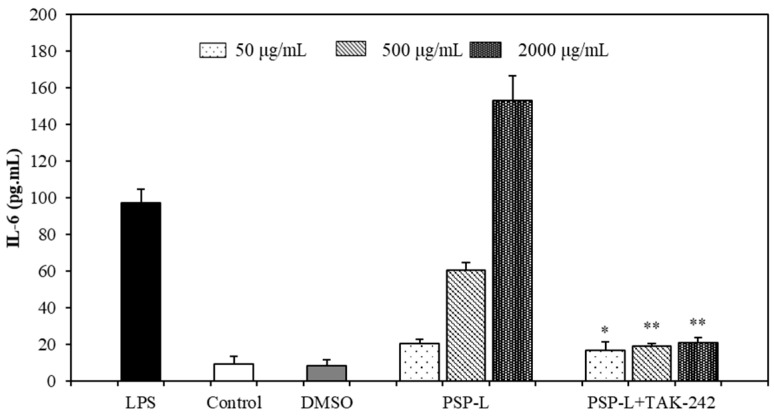
Effect of TLR4 inhibitor on PSP-L induced IL-6 production in RAW264.7. The data shown are means ± SD (*n* = 3). All data were analyzed statistically using a one-way analysis of variance. (*) *p* < 0.05 and (**) *p* < 0.01, compared with the PSP-L group, respectively.

**Figure 9 marinedrugs-18-00538-f009:**
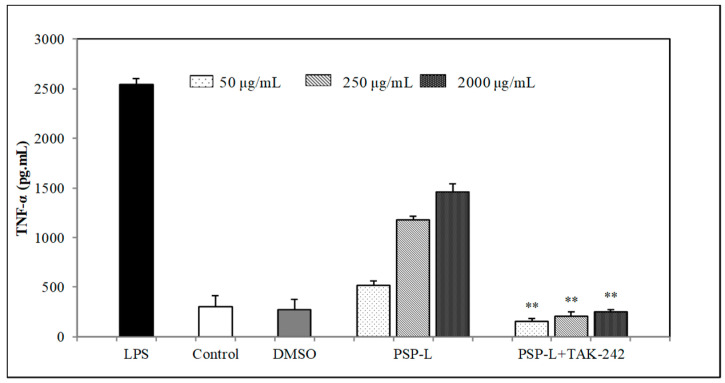
Effect of TLR4 inhibitor on PSP-L induced TNF-α production in RAW264.7. The data shown are means ± SD (*n* = 3). All data were analyzed statistically using a one-way analysis of variance. (**) *p* < 0.01, compared with the PSP-L group, respectively.

**Figure 10 marinedrugs-18-00538-f010:**
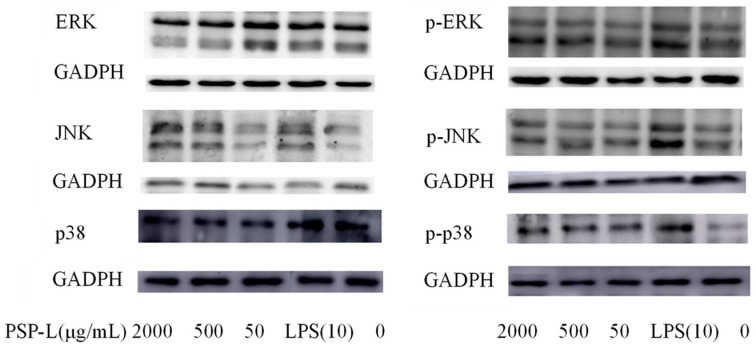
Effect of PSP-L (50, 500, 2000 μg/mL) on MAPK activation in RAW 264.7 cells. The result shown are means ± SD (*n* = 3), as determined from triplicate experiments.

**Figure 11 marinedrugs-18-00538-f011:**
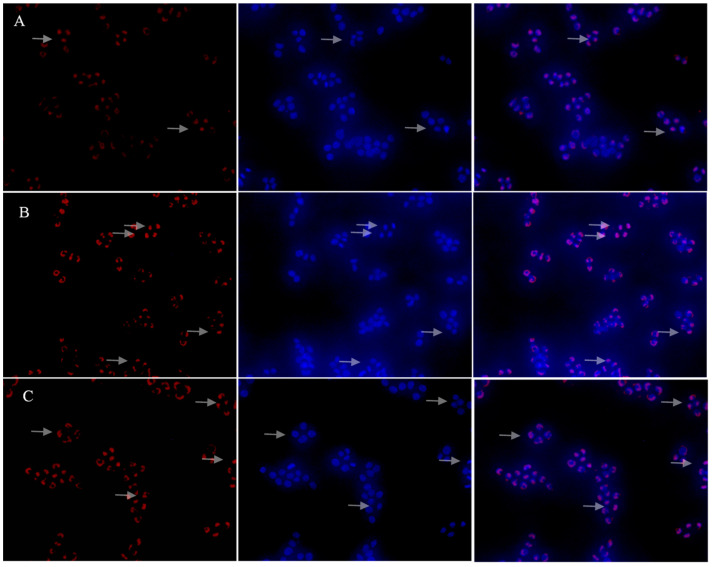
Effects of PSP-L on the activation of the NF-κB signaling pathway in RAW 264.7 cells. (**A**) showed the control group, (**B**) showed LPS (10 μg/mL) positive control group, (**C**) showed PSP-L 2000 (μg/mL) treated group.

**Figure 12 marinedrugs-18-00538-f012:**
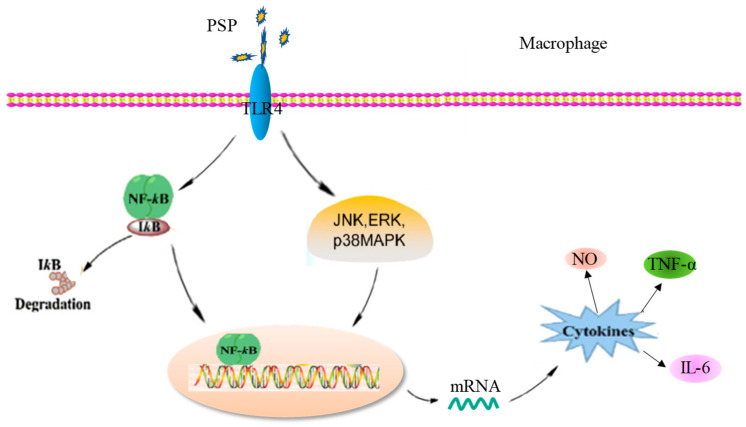
Possible mechanism of PSP activating RAW 264.7 cells.

**Figure 13 marinedrugs-18-00538-f013:**
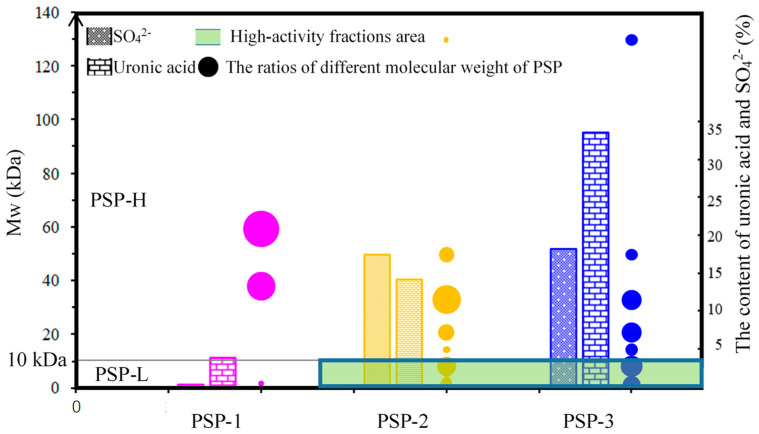
Distribution of high immunostimulatory activity fractions of PSP. Note: PSP-L here denotes the molecular weight under 10 kDa, PSP-H here denotes the molecular weight above 10 kDa. Pink represents PSP-1, orange represents PSP-2 and blue represents PSP-3. The green area represents high-activity PSP fractions.

**Figure 14 marinedrugs-18-00538-f014:**
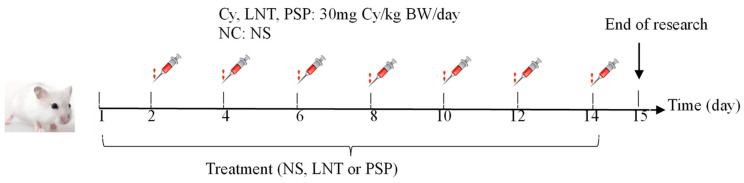
Modeling and treatment of Cy immunodeficiency mice.

**Table 1 marinedrugs-18-00538-t001:** Effect of PSP on organ index and the number of PWBC and PBL in mice. Values are represented as the means ± SD as determined from 6 mice.

Group	Spleen Index	Thymus Index	PWBC (×10^9^/L)	PBL (×10^9^/L)
NC	2.97 ± 0.23	2.54 ± 0.2	4.04 ± 0.75	3.25 ± 0.38
Cy	1.86 ± 0.15 **	1.11 ± 0.14 **	1.47 ± 0.42 **	1.05 ± 0.17 **
LNT	2.46 ± 0.3 *^,#^	1.87 ± 0.27 **^,#^	2.38 ± 0.54 **^,#^	1.84 ± 0.32 **^,##^
PSP_L_	2.02 ± 0.08 **	1.18 ± 0.34 **	1.96 ± 0.69 **	1.51 ± 0.34 **^,#^
PSP_M_	2.16 ± 0.17 *^,#^	1.38 ± 0.27 **	2.06 ± 0.641 **^,#^	1.65 ± 0.13 **^,#^
PSP_H_	2.26 ± 0.06 *^,#^	1.55 ± 0.22 *^,#^	2.27 ± 0.47 **^,##^	1.79 ± 0.48 **^,#^

Note: PSP_L_ denotes low dose PSP (500 mg/kg BW/day), PSP_M_ denotes middle dose PSP (1000 mg/kg BW/day), PSP_H_ denotes high dose PSP (1500 mg/kg BW/day), (*****) *p* < 0.05 and (******) *p* < 0.01 vs. NC group; # *p* < 0.05 and ## *p* < 0.01 vs. Cy group.

**Table 2 marinedrugs-18-00538-t002:** The protocols for immunosuppression induction and treatment.

Group	Cy-Induced (mg/kg/day) (Day 2, 4, 6……14)	Drug and Dosage (mg/kg) Once a Day
NC	normal saline	NS
Cy	30 mg Cy/kg BW/day	NS
LNT	30 mg Cy/kg BW/day	500 mg LNT/kg BW/day
PSP_L_	30 mg Cy/kg BW/day	500 mg PSP/kg BW/day
PSP_M_	30 mg Cy/kg BW/day	1000 mg PSP/kg BW/day
PSP_H_	30 mg Cy/kg BW/day	1500 mg PSP/kg BW/day

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
