# Peer review of "Immunostimulatory Effects of Polysaccharides from Spirulina platensis In Vivo and Vitro and Their Activation Mechanism on RAW246.7 Macrophages"

_marinedrugs, 2020, doi:10.3390/md18110538_

Round 1

Reviewer 1 Report

After reading carefully the revised manuscript, I have seen an improvement, however I think it is not ready to be published yet.

There are many mistakes and an extensive editing of English language and style are highly required.

Some of my suggestions have been accepted, and supposed to have been modified as authors said, but in many cases they still remain the same.

In addition, some general issues still need to be addressed:

  • All the description of each abbreviation must be included the first time that is writen.
  • Now I have understood what PSP L, M and H are, but this is described in M&M section, and since results appear first, it is imposible to know the concentrations while reading the paper. This should be explained in the Results section too.
  • Sometimes “friction” instead of “fraction” is writen. Please revise it.

Regarding some comments, some issues still need to be addressed/claryfied:

  • Comment 13: I cannot find where the “n” has been included. Please revise it.
  • Comment 15: I had previously recommended to include this sentence in discussion and this has not been performed.
  • Comment 20: I cannot see any modification in the text. Is it correct?
  • Comment 22: I cannot see anything related to an Ethics Committee. Is it correct?
  • Comment 32: “anti-bodies” has not been corrected.
  • Comment 35: In line 426, I should say “but 425 more controls and another experiments are needed to be able to assume the involvement of TLR4 426 in the recognition of PSP-L”, since they have not been performed.

Reviewer 2 Report

The authors have improved the manuscript appropriately according to the suggestions from the reviewers. The manuscript can be accepted in the current form.

Author Response

Thank you!

This manuscript is a resubmission of an earlier submission. The following is a list of the peer review reports and author responses from that submission.

Round 1

Reviewer 1 Report

The article concerns immunostimulatory action of polysaccharides from the green alga Spirulina platensis and its mechanism of activation of RAW246.7 macrophages. The polysaccharide (PSP), that is as a complex multi-component system, has been isolated and separated on three polysaccharide fractions (PSP-1, PSP-2 and PSP-3) with different acidic groups were obtained after PSP was fractionated by the DEAE-52 cellulose chromatography, and two polysaccharide fractions (PSP-L and PSP-H) with different molecular weight were obtained by ultrafiltration centrifugation. The chemoprotective effects of the PSP in cyclophosphamide (Cy) treated mice, namely restoring of cell immunity of threated mice were found. The acidic polysaccharide fractions of PSP-2, PSP-3 and PSP-L with small molecular weight had the higher immunostimulatory activity. Signaling pathway research results indicated that PSP-L activated RAW264.7 macrophage cells through MAPKs, NF-κB signaling pathways via TLR4 receptor.

The article is good written, the results of bioassay and studying of biological action mechanism seems to be reliable. The bioassay part of the article is excellent but the absence of any chemical logic in this article is the main flaw of the manuscript.

The authors even didn’t attempt to separate the total polysaccharide fraction (PSP) on homogenous components. They independently separate the fraction by acidity and by molecular weight but didn’t separate the fractions with different acidity by molecular weight or opposite. It should be fixed, of course, and homogenous components should be characterized by monosaccharide content, sulfate groups content, uronic acid content etc. The biological activity should be reinvestigated on homogenous components also. It should be enough for this article but the chemical structure of homogenous polysaccharides also should be studied in future investigations.

Because the PSP should be separated on homogenous components both by acidity and molecular weight and their activity also should be reinvestigated, this manuscript should be rejected and resubmitted after normal separation of the PSP and reinvestigation of homogenous fractions.

The references are not correctly formatted: the year should be in bold, the volume – in italic, the page numbering full in all the references, the names of the journals should be shortened etc. The word “macrophages” should be added after “RAW246.7” in the title.

Reviewer 2 Report

The work entitled “Immunostimulatory effects of polysaccharides from Spirulina platensis in vivo and in vitro and its mechanism of RAW246.7” describes the effect different fractions extrated from Spirulina platensis can have reverting the immunossupression induced by cysplatine in vivo, and the immunostumulating effect they can induce on macrophages in vitro. It is worth it to mention the hard work the authors have carried out, however several and important methodological, text editing and organization mistakes must be solved in order to be accepted for publication.

In this sense, some general issues must be addresed:

  • English must be importantly improved. I have tried to correct some mistakes, but I would strongly recommend the text to be revised by a native English person.
  • A hard text editing review must be done, since there are several sentences not well formulated or unfinished, which makes it imposible to be able to understand some content.
  • In some cases, different Font style can be detected.
  • There is a lot of information missing in the Material and Methods section (described below). And one of the most important is the number of repetition every experiment has been performed, and also the number of replicates in each experiment.
  • It is not clear why in vivo, only fractions PSP-L and H are administered, but not 1, 2 and 3? And what about PSP-M? It just only appears in some results. This must be clarified.
  • In vitro and in vivo must be written in italics.
  • The discussion section must be improved. There are many mispelled sentences.
  • I would suggest to separate both, materials and methods, and results, in In vitro and In vivo It would help a lot to the reader.

Some details must be adressed too:

  • In line 20: “can” is not writen in the correct verbal tense
  • In line 22: “enhanced” is not writen in the correct verbal tense
  • Paragraph in lines 48-51 has many mistakes. This must be revised.
  • Paragraph in lines 52-58 has some mistakes. This must be revised.
  • Sentence between 62-63 is mispelled. It need to be revised.
  • In line 65: “LNT-treated”. Authors should define whats is LNT and explain why they use it as a control.
  • In lines 72-76 is stated that PWBC and PBL are elevated in a dose-dependent manner, however this data is not represented. I would suggest to indicate that this data is not shown if this is the case.
  • Regarding this result, I would like to know how B lymphocyte (PBL, called by the authors) could be differentiated from T lymphocytes. Have the authors performed flow cytometry based on different surface markers? As far as I know, conventional hematology analyzer are not able to differentiate them. This should also be included in M and M section.
  • All the figure captions must include “n”, meaning how many times the experiment has been performed or how many mice samples were used to represent the different graphics.
  • Sentence in lines 108-109 is misspelled. In addition, the increase of these analytes levels over LPS, only can be observed with NO and IL-6, but not with TNF-alpha, and also this does not seem to happen with all the fractions of PSP. Ths must be revised and corrected.
  • In figure 4 caption, p<01 is repeated.
  • In line 119, the title 2.6 should be reformulated, it is incomprehensible.
  • Sentence in lines 120-122 should be revised, it has many mistakes.
  • In line 124, what is SGRP1?
  • Regarding experiment using the TLR4 inhibitor (section 2.6), many controls are missing. LPS and control cells should have also been treated with the inhibitor, otherwise it is not reliable to believe the effect on PSP-treated cells. In addition, how long these cells were incubated with different treatments? This data must be included.
  • Depending on the time of exposure in section 2.6, the concentration of cytokines in these experiments are comparable to that included previously in the section 2.5? If this is the case, it could be worth it to mention in the text.
  • In line 137, it is stated that there is a “significantly increase”, however, only one experiment is represented. Once again, it is needed to inlcude the “n”, the numer of experiment repetitions.
  • The sentence between 137-140 should be included in the discussion section rather than in results.
  • Paragraph between lines 145-151 has some mistakes. This must be revised.
  • In figure 10, the caption is describing a result rather than the caption per se, this should be corrected. I would also suggest to include in the figure which colum correspond to each fluorochrome/merge, and which line correspond to each treatment.
  • Sentence between lines 155-156 must be rewritten.
  • In line 167, it is stated that “Immune function and immune prognosis are reflected well by the thymus and spleen indices because these organs play important roles in nonspecific immunity”. And there is an important mistake, since thymus is a very important organ for specific immunity. This must be corrected.
  • Sentence in lines 185-187 is not well written, it sems that are results from a work performed by the authors rather than an alrealy known fact. This should be corrected.
  • Sentences between 197-201 should be rewritten.
  • In line 201, it is stated again that TNF-alpha levels in cells trated with PSP are higher than LPS treated ones, and this cannot be observed in the graphics. This must be corrected or explained.
  • Sentence in lines 217-218 must be rewritten.
  • Paragraph between lines 225-235 and the corresponding figure 12 explanation, must be improved, specially describing what high-activity frictions area are. And it also would be helpful to refer to results in supplmentary information. It is very difficult to understand this section.
  • Regarding Materials and Methods, there are several aspects that must be improved or included:
    • Regarding the experimental work with mice, is there a code number related to the approvement by the Ethics Committee? This must be included in the text.
    • In table 2, which fraction is PSPm?
    • In section 4.6, the title is “Peripheral white blood cell, red blood cell and platelets counts”, however, only data regarding WBC and BL are represented. This should be corrected. And in addition, and as it was mentioned before, how the authors cuantified B cells? The name of the hematology analyzer must be included.
    • In 4.7 section, which was the frequency for cells to be Split?
    • In 4.8 section authors sometimes use “absortion” instead of the correct one “absorbance”. This must me corrected.
    • In sections 4.8 and 4.9 the assays refer to prolilferation and cell viability, however, as I can understand after reading the results section, the kit used is the same? This must be clarified and homogenize.
    • In section 4.10, authors use 1x106 cells per well in a 96 wells plate. Taking into account that for the cell viability assay they have used lower amount of cells and my own experience, these are a lot of Raw 264.7 cells to be cultured for a long period, have the authors checked the viability before analyzing the pinocytic capability?
    • Regarding the number of cells, authors use different number of cells in different assays despite the type of plate is the same, however, the concentration of each treatment remains the same. This would be translated in a different “cell nº/amount of treatment” ratio. Have the authors took this into account?
    • In section 4.10, in line 326, it should be “cell lysis solution” instead of “cell lysate”.
    • In section 4.11 only PSP-L is mentioned, but in results all the fractions are represented.
    • In section 4.12, there are some mistakes or some information is missing, and this must be corrected:
      • Sentence in line 343
      • In line 345: It should be “treatment” instead of “treated”?
      • How long are cells in contact with different treatments?
      • Have the authors included control cells (negative control and LPS-stimulated control) with and without the TLR4 inhibitor? As it was mentioned previously, these controls must be included in order to confirm the role of this receptor in the recognition of PSP.
    • In section 4.13, in line 359, it should be “antibodies” instead of “anti-bodies”
    • In section 4.14, which plates have been used? What fixative medium has been used? Which fluorescent inverted microscope has been used? This information must be included.
    • In section 4.15, there are some mistakes. The paragraph must be revised and corrected.
  • In conclusion section:
    • In line 378, it should be “The present study” instead of “This present study”?
    • In line 381, it should be “restore” instead of “restored”?
    • Regarding the sentence in lines 382-383 “The in vitro assays proved its immunomodulatory activity and indicated that its fraction PSP-L 382 activated RAW264.7 cells through MAPKs, NF-κB signaling pathways via TLR4 receptor”, as it was previously commented, more controls and another experiments are needed to be able to asume the involvement of TLR4 in the recognition of PSP. This sentence must be reformulated.
  • In supplementary information:
    • In section S1, why PSP-H and L graphics are not included?

Reviewer 3 Report

The paper by Wu, X. is interesting and it is really having a lot of data to support the finding. I have a couple of comments to improve the manuscript:

(1) Why the author has selected macrophage RAW246.7 cell line to do this study? I understand that this study is related to the TLR4-signaling pathway, but why not some cells from the adaptive immune system especially B-cells was considered?

(2) In addition, it is also good for the understanding of the paper how polysaccharides have inflammatory effects on macrophages and other immune cells. Thus, the introduction part should be improved.